# STEP: Segmenting and Tracking Every Pixel

**Mark Weber**[1][*]    **Jun Xie**[2]    **Maxwell Collins**[2]    **Yukun Zhu**[2]    **Paul Voigtlaender**[2,3]

**Hartwig Adam**[2]    **Bradley Green**[2]    **Andreas Geiger**[4]    **Bastian Leibe**[3]

**Daniel Cremers**[1]    **Aljoša Ošep**[1]    **Laura Leal-Taixé**[1]    **Liang-Chieh Chen**[2]

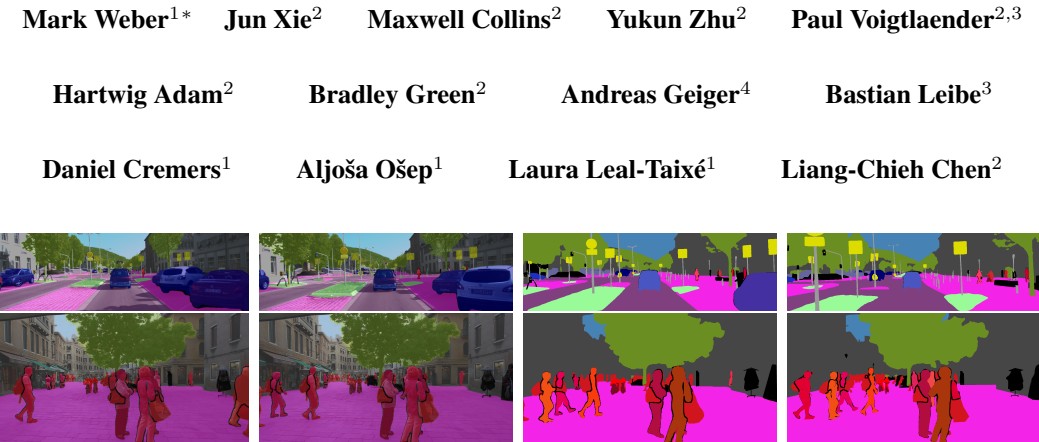

Figure 1: Our proposed ground-truth labels of KITTI-STEP (top) and MOTChallenge-STEP (bottom).

## Abstract

The task of assigning semantic classes and track identities to every pixel in a video is called video panoptic segmentation. Our work is the first that targets this task in a real-world setting requiring dense interpretation in both spatial *and* temporal domains. As the ground-truth for this task is difficult and expensive to obtain, existing datasets are either constructed synthetically or only sparsely annotated within short video clips. To overcome this, we introduce a new benchmark encompassing two datasets, KITTI-STEP, and MOTChallenge-STEP. The datasets contain long video sequences, providing challenging examples and a test-bed for studying long-term pixel-precise segmentation and tracking under real-world conditions. We further propose a novel evaluation metric Segmentation and Tracking Quality (STQ) that fairly balances semantic and tracking aspects of this task and is more appropriate for evaluating sequences of arbitrary length. Finally, we provide several baselines to evaluate the status of existing methods on this new challenging dataset. We have made our datasets, metric, benchmark servers, and baselines publicly available, and hope this will inspire future research.

## 1  Introduction

Dense and pixel-precise video scene understanding is of fundamental importance for applications such as autonomous driving, film editing and spatio-temporal reasoning. More specifically, while the semantic interpretation helps with tasks such as estimating the drivable area for an autonomous vehicle, the tracking of objects enables us to anticipate the temporal evolution of the surroundings, which is critical for motion planning and obstacle avoidance.

**Challenges.** Moving towards this goal, there are three challenges that we find not addressed by previous benchmarks. First, the ability of *explaining every pixel* of a continuous input from cameras. Second, changes in the input signal over time can happen quickly and hence, demanding that *we evaluate with the same high-frequency* as that of the changes that occur. Third, interpretation of continuous sensory input requires temporally consistent scene understanding, *i.e.*, *long-term tracking*,

---

[*]This work was partially done during an internship. [1]Technical University Munich, [2]Google Research, [3]RWTH Aachen University, [4]MPI-IS and University of Tübingen

35th Conference on Neural Information Processing Systems (NeurIPS 2021) Track on Datasets and Benchmarks.

which current benchmarks and metrics are not suitable for. The aim of this work is to advance this field by introducing a suitable benchmark and metric.

In the past, image benchmarks such as PASCAL VOC [21], ImageNet [52], and COCO [40] played a pivotal role in the astonishing progress of computer vision research over the last decade, allowing the community to evaluate different methods in a standardized way. Real-world datasets for various tasks [38, 57, 54, 49, 25, 13] were used to fairly measure the progress and highlight key innovations.

For the purpose of holistic image understanding, Kirillov *et al*. [36] introduced the concept of *panoptic segmentation* as a combination of semantic segmentation and instance segmentation. Kim *et al*. [34] subsequently introduced the notion of video panoptic segmentation (VPS). Yet, they merely label a sparse subset of pixels from short real-world video snippets that are not suitable for dense pixel-precise video understanding. Moreover, existing *synthetic* datasets [29, 34] struggle to evaluate performance in the real-world due to the domain shift [53].

For evaluation of VPS, existing metrics [29, 34] build upon metrics for panoptic segmentation and multi-object tracking. Since a metric can be significant in deciding the community's research direction, biases in the metric can hinder promising innovations.

**Contributions.** The contribution of this work is threefold:

(1) We introduce more suitable benchmark datasets that in particular allow spatio-temporally dense and pixel-centric evaluation. Our proposed benchmark extends the existing KITTI-MOTS, and MOTS-Challenge datasets [61] with spatially and temporally dense annotations. We seek to label *every* pixel with a semantic class and track ID. As in panoptic segmentation [36], we treat each non-countable region, such as the *sky*, as belonging to a single track. For the most salient countable classes, each instance is assigned a semantic class and a unique ID throughout the video sequences.

(2) After studying prior metrics in detail in Sec. 4, we propose the Segmentation and Tracking Quality (STQ) metric that is more suitable to access the segmentation and tracking performance of algorithms. STQ is defined at the pixel level and provides an accurate and intuitive comparison against the ground-truth at a fine-grained level. The core principle of our benchmark is that each pixel in each frame matters when evaluating an algorithm.

(3) Finally, our datasets and metric provide us a valid test-bed for evaluating several baselines that show the effect of unified *vs*. separate (*c.f*. Sec. 5) and motion- *vs*. appearance-based methods on our benchmark. This includes methods that use optical flow for mask propagation [47, 44, 58] or methods inspired from state-of-the-art tracking work [4, 67]. Test servers will enable a fair benchmark of methods. This provides a complete framework to enable research into dense video understanding, where both segmentation and tracking are evaluated in a detailed and holistic way. In summary,

- We present the first real-world spatially and temporally dense annotated datasets KITTI-STEP and MOTChallenge-STEP, providing challenging segmentation and (long) tracking scenes (Sec. 3).
- We analyze in-depth the recently proposed metrics [34, 29], and based on our findings propose the Segmentation and Tracking Quality (STQ) metric (Sec. 4).
- We showcase simple baselines based on established segmentation and tracking paradigms, motivating future research in end-to-end models (Sec. 5, Sec. 6).

## 2 Related Work

**Panoptic Segmentation.** The task of panoptic segmentation combines semantic segmentation [27] and instance segmentation [24], and requires assigning a class label and instance ID to all pixels in an image. It is growing in popularity, thanks to the proposed benchmark [36]. The corresponding panoptic quality (PQ) metric is the product of recognition quality and segmentation quality.

**Video Semantic Segmentation.** Compared to image semantic segmentation [21, 42, 10], video semantic segmentation [69] is less common in the literature, possibly due to the lack of benchmarks. Video semantic segmentation differs from our setting in that it does not require discriminating different instances and hence, also no explicit tracking.

**Multi-Object Tracking.** The task of multi-object tracking (MOT) is to accurately track multiple objects by associating bounding boxes in videos [7, 4, 48]. Focusing on tracking, this task does not require any segmentation. The MOT-Challenge [19, 20] and KITTI-MOT [23] datasets are among the most popular benchmarks, and the tracking performance is measured by the CLEAR MOT metrics [5] along with a set of track quality measures introduced in [63]. Recently, [43] propose HOTA (Higher Order Tracking Accuracy), which explicitly balances tracking results attributed from accurate detection and association.

**Video Instance Segmentation.** Combining instance segmentation and MOT, the goal of video instance segmentation (VIS) [65] is to track instance masks across video frames. This task is also known as multi-object tracking and segmentation (MOTS) [61]. Yet, the respective work adapt different evaluation metrics, focusing on different perspectives of the task. The prominent Youtube-VIS dataset [65] focuses on short sequences (3-6 seconds, every 5th frame annotated) in which objects are present from start to end. For that, $AP^{mask}$ [24, 40] is adapted to videos by extending the IoU to the temporal domain (3D IoU). The focus of MOTS datasets, KITTI-MOTS, and MOTS-Challenge [61], is on more general scenarios with appearing and disappearing objects within long videos. Therefore, MOTSA (Multi-Object Tracking and Segmentation Accuracy) is used, the mask-based variants of the CLEAR MOT metrics [5]. The two MOTS datasets focus on challenging urban sequences like one encounters in an autonomous driving setting. In contrast to our task, these benchmarks do not consider non-instance regions and hence, do not require pixel-precise video understanding. We build on top of these datasets by adding *semantic segmentation* annotations to obtain KITTI-STEP and MOTChallenge-STEP.

**Video Panoptic Segmentation.** Recently, panoptic segmentation has also been extended to the video domain. Video Panoptic Segmentation (VPS) [34] requires generating the instance tracking IDs along with panoptic segmentation results across video frames. Previous datasets [34, 29] and corresponding metrics fail to properly evaluate scenarios that require both short- and long-term segmentation and tracking. Specifically, due to expensive annotation efforts for VPS datasets, most of the reported experimental results in the literature [34, 29] are conducted on synthetic datasets [50, 22], making it hard to generalize to real-world applications [28, 59]. Exceptionally, [34] annotates Cityscapes video sequences [17] for 6 frames out of each short 30 frame clip (about 1.8 seconds), hence focusing on short-term tracks. Both benchmarks [34, 29] evaluate with different metrics, VPQ (Video Panoptic Quality) and PTQ (Panoptic Tracking Quality), respectively. VPQ, tailored for sparse annotations on short clips, was not designed for evaluating long-term tracking, while PTQ penalizes recovery from association errors and may produce hard to interpret negative evaluation results. Motivated by the issues above, we propose two datasets densely annotated in space and time, KITTI-STEP and MOTChallenge-STEP requiring *long-term* segmentation and tracking. Additionally, we propose a novel STQ metric that gives equal importance to segmentation and tracking.

## 3 Datasets

**Overview.** We collect two densely annotated datasets in both spatial and temporal domains, *i.e.*, every pixel in every frame is annotated, building on top of KITTI-MOTS and MOTS-Challenge [61]. Both have carefully annotated tracking IDs for 'pedestrians' and 'cars' (KITTI-MOTS only) in real-world scenes, as they are the most salient moving objects. So far, other common semantic classes in urban scenes, such as 'bicycles' and 'road', have all been grouped into one 'background' class, impeding pixel-level scene understanding. Therefore, our additional *semantic segmentation* annotation, defined by the 19 Cityscapes classes [17], enriches the KITTI-MOTS and MOTS-Challenge datasets. Specifically, all 'background' regions are carefully annotated, and classes that are not tracked, *i.e.*, everything except 'pedestrians' and 'cars' in our case, are considered as a single track, similar to the 'stuff' definition in panoptic segmentation [36]). The resulting datasets, called KITTI-STEP and MOTChallenge-STEP, present challenging videos requiring long-term consistency in segmentation and tracking under real-world scenarios. Fig. 1 shows ground-truth annotations from the proposed dataset.

**Semi-automatic annotation.** Similar to [8, 2, 61], we collect our annotations in a semi-automatic manner. In particular, we employ the state-of-the-art Panoptic-DeepLab [15, 9], pretrained on the Mapillary Vistas [46] and Cityscapes [17] datasets to generate pseudo *semantic* labels for each frame.

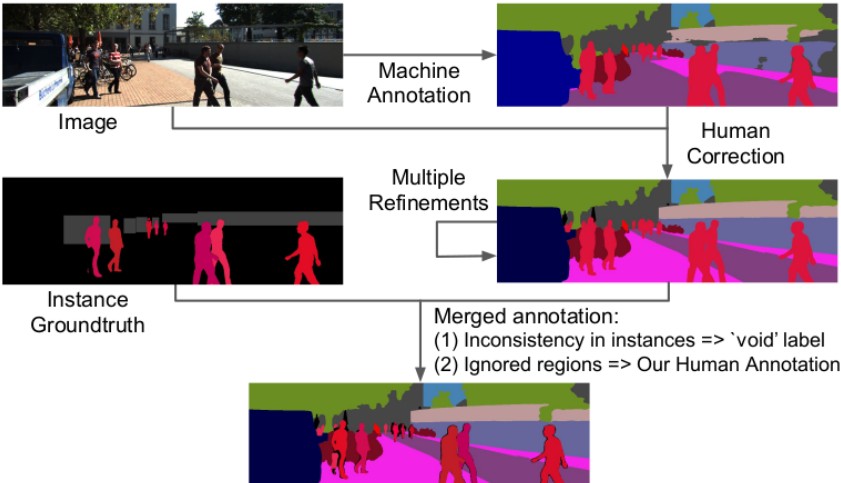

Figure 2: Annotation process: The machine annotation *semantic segmentation* from Panoptic-DeepLab is corrected by human annotators with multiple refinements. The resulting annotation is further merged with the existing instance ground-truth from KITTI-MOTS and MOTS-Challenge.

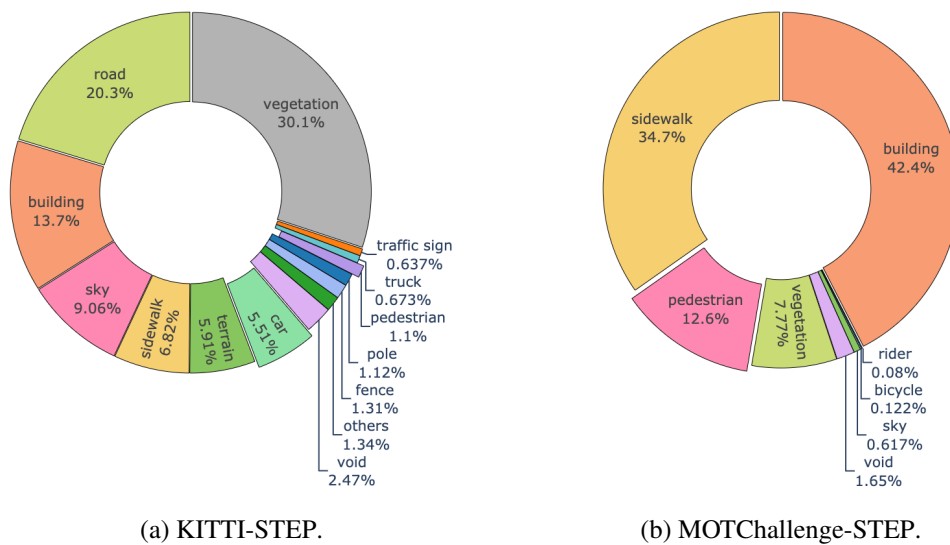

(a) KITTI-STEP.                    (b) MOTChallenge-STEP.

Figure 3: Label distribution in KITTI-STEP and MOTChallenge-STEP.

The predicted *semantic segmentation* is then carefully refined by human annotators. On average, it takes human annotators around 10 minutes to annotate each of the more than 20000 frames. The refining process is iterated twice to guarantee high-quality per-frame annotations and consistent labels across consecutive video frames. We illustrate our semi-automatic annotation process in Fig. 2 and provide all technical details regarding the merging procedure in the supplement. It is evident from the illustration, that human refinements and corrections are required to achieve high quality annotations.

**KITTI-STEP dataset.**[2]  KITTI-STEP has the same train and test sequences as KITTI-MOTS (*i.e.*, 21 and 29 sequences for training and testing, respectively). Similarly, the training sequences are further split into training set (12 sequences) and validation set (9 sequences). This dataset contains videos recorded by a camera mounted on a driving car. Since the sequence length sometimes reaches over 1000 frames, long-term tracking is required. Prior work [60] has shown that with increasing sequence length, the tracking difficulty increases too. These sequences involve regularly (re-)appearing and disappearing objects, occlusions, light condition changes and scenes of pedestrian crowds. The semantic label distribution is shown in Fig. 3.

---

[2]http://cvlibs.net/datasets/kitti/eval_step.php

| Dataset statistics | City-VPS [34] | KITTI-STEP | MOTChallenge-STEP |
|---|---|---|---|
| # Sequences (trainval/test) | 450 / 50 | 21 / 29 | 2 / 2 |
| # Frames (trainval/test) | 2,700 / 300 | 8,008 / 11,095 | 1,125 / 950 |
| # Semantic classes | 19 | 19 | 7 |
| # Annotated Masks† | 72,171 | 126,529 | 17,232 |
| Every frame annotated | ✗ | ✓ | ✓ |
| Annotated frame rate (FPS) | 3.4 | 10 | 30 |
| | Max/Mean/Min | Max/Mean/Min | Max/Mean/Min |
| Annotated frames per seq.† | 6 / 6 / 6 | 1,059 / 381 / 78 | 600 / 562 / 525 |
| Track length (frames)† | 6 / 3 / 1 | 643 / 51 / 1 | 569 / 187 / 1 |

(a) Track length distributions of KITTI-STEP. (b) Real-world dataset comparison. † refers to the trainval set.

Figure 4: Dataset statistics, comparison and track length distribution of KITTI-STEP.

**MOTChallenge-STEP dataset.**[3]    Four sequences are annotated for MOTChallenge-STEP. In particular, we split these sequences into two for training and two for testing. This dataset contains only 7 semantic classes, as not all of Cityscapes' 19 semantic classes are present. Even though MOTChallenge-STEP contains only one tracking class, we argue that tracking is rather difficult. First, the difficulty of tracking increases with the number of simultaneously, visually similar objects present in a scene. At the same time, this increases the chance of occlusions to occur. Concurrent work [33] has shown that this is indeed the case in the crowded MOTChallenge-STEP sequences. Second, the limited number of sequences makes training harder. We pose this as an additional challenge for future work to overcome the notion of requiring large amounts of annotated data by making use of weakly-, semi- and unsupervised approaches to improve upon our baselines. We visualize the semantic label distribution in Fig. 3.

**Dataset comparison.**  In Tab. 4b, we compare KITTI-STEP and MOTChallenge-STEP with the only non-synthetic VPS dataset, Cityscapes-VPS [34]. Notably, we have more than 6 times the number of annotated frames and our longest video sequence is 176 times longer. Thus, our datasets require long-term tracking instead of focusing on short clips. KITTI-STEP contains sequences *of over 1000 annotated frames* that are *all* used to evaluate the performance. Hence, our datasets are not biased towards segmentation, as is the case when evaluating only on *6 annotated frames*. Additionally, our pixel-dense annotations in space and time enable methods to work directly on video clips.

**Tracking Difficulty.**  As shown by [60, 33], tracking difficulty primarily increases with number of objects, occlusions and tracking length. The track length distribution in Fig. 4a shows the importance of long-term tracking and gives evidence that evaluating on a few frames is not representative of real-world scenarios. An extended discussing about tracking difficulty can be found in the supplement. In short, close to half of all tracks in [34] are only present in a single frame requiring no tracking. Hence, with the available annotation budget we decided to follow the best-practices in tracking benchmarks of focusing on the most salient classes as more classes does not mean harder tracking. The remaining object classes account for less than 2% of all pixels in the dataset. Therefore, our datasets present a more challenging and practical scenario, requiring both short- and long-term tracking.

**Synthetic datasets.**  Prior work [34] has used the synthetic dataset VIPER [50], which contains video game footage from the 2013 released game GTA5. [51] has studied the realism of VIPER. Even when showing VIPER and Cityscapes images only for 500ms, the real-world data was consistently ranked to be more realistic. Moreover, [53] has shown a large domain shift between synthetic GTA5 data and Cityscapes. Therefore, synthetic data has a purpose for pretraining, but relying purely on it does not work well enough for real-world applications, motivating the need for new benchmarks.

# 4   Metric

A detailed analysis of existing metrics is essential for developing a new one. We first look into basic components PQ [36] and MOTSA [61], before diving into the VPQ [34] and PTQ [29] metrics. We then condense these insights into properties that a good STEP metric should satisfy.

---

[3]https://motchallenge.net/data/STEP-ICCV21/

## 4.1 Metric Analysis

**Panoptic Quality (PQ).** Kirillov *et al.* [36] proposed PQ to measure panoptic segmentation results. The set of *true positive* ($TP_c$), *false positive* ($FP_c$) and *false negatives* ($FN_c$) segments, for a particular semantic class $c$, is formed by matching predictions $p$ to ground-truth $g$ based on the IoU scores. A threshold of greater than 0.5 IoU is chosen to guarantee unique matching. The overall metric is

$$PQ_c = \frac{\sum_{(p,g) \in TP_c} IoU(p,g)}{|TP_c| + \frac{1}{2}|FP_c| + \frac{1}{2}|FN_c|}. \tag{1}$$

Defining a predicted segment with IoU of 0.51 as true positive and a segment with IoU of 0.49 as false positive (and the ground-truth segment as false negative) does not align well with human perception. Moreover, the metric is highly sensitive to false positives that cover few pixels, which is why *most, if not all,* panoptic segmentation methods use an additional post-processing step to remove *s*tuff classes with few pixels or add 'void' class predictions [64] to increase the PQ score.

$\Rightarrow$ A metric should be insensitive to such post-processing, and consider all predictions without requiring a threshold to define true positives on a segment level.

**MOTA/MOTSA.** Multi-Object Tracking Accuracy (MOTA) [5] introduced the concept of ID switches $IDSW_c$ between frames of the prediction w.r.t. the ground-truth $GT_c$. It became the standard in many tracking benchmarks [23, 45]. Its derivative Multi-Object Tracking and Segmentation Accuracy (MOTSA) [61] additionally considers segmentation, by matching objects based on segment overlap:

$$MOTSA_c = \frac{|TP_c| - |FP_c| - |IDSW_c|}{|GT_c|}. \tag{2}$$

MOTA and MOTSA were analyzed exhaustively by [43]. The most relevant drawbacks are:

- D1. The metric penalizes ID recovery, *i.e.*, correcting mistakes gives worse scores.
- D2. The metric is unbounded and can take negative values, making scores hard to interpret.
- D3. The tracking precision is not considered, only recall is (*c.f.* [43] for details).
- D4. For detection, precision is much more important than recall, leading to a high imbalance.
- D5. Similar to PQ, MOTSA uses the threshold-based matching to define the set of true positives.

$\Rightarrow$ A metric should consider both precision and recall, and should not penalize correcting mistakes.

**VPQ.** Video Panoptic Quality (VPQ) [34] for video panoptic segmentation is based on PQ and tailored to the sparsely annotated Cityscapes-VPS dataset. VPQ computes the average quality w.r.t. single frames and small spans of frames, by using a *2D and 3D IoU* for matching, respectively:

$$VPQ = \frac{1}{K} \sum_k \frac{1}{N_{classes}} \sum_c \frac{\sum_{(p,g) \in TP_c^k} IoU_{2D/3D}(p,g)}{|TP_c^k| + \frac{1}{2}|FP_c^k| + \frac{1}{2}|FN_c^k|} \tag{3}$$

We note that these spans are constructed from every 5th video frame. Considering this averaging and that at most $K = 4$ frames are taken into account, the metric puts much more emphasis on segmentation than on association. As noted in [34], when using more than 4 frames, the difficulty of the 3D IoU matching increases significantly. We agree with that assessment, but argue that we should by-pass threshold-based matching and propose a metric defined on the *pixel* level. When considering a setting like ours in which there are more *stuff* classes than countable *things* classes, VPQ reduces this task almost to video semantic segmentation as the metric is averaged over all classes. As a consequence, the importance of association varies with the dataset class split into *stuff* and *things*.

$\Rightarrow$ A metric should be suitable for evaluating not only short-term but also long-term tracks as well as give equal importance to segmentation and tracking, as required for real-world scenarios.

**(s)PTQ.** Hurtado *et al.* propose the Panoptic Tracking Quality (PTQ) measure [29]:

$$PTQ_c = \frac{\sum_{(p,g) \in TP_c} IoU_{2D}(p,g) - |IDSW_c|}{|TP_c| + \frac{1}{2}|FP_c| + \frac{1}{2}|FN_c|}. \tag{4}$$

The metric combines PQ and MOTSA, by changing the numerator of PQ to subtract the number of ID switches $IDSW_c$. The metric is computed in *2D on a frame by frame basis*, thus is incapable of looking beyond *first-order association* [43] mistakes. Also, PQ's original threshold-based matching is applied. Following this design, PTQ inherits the drawbacks from PQ as well as most issues from MOTSA as described above.

$\Rightarrow$ A metric should evaluate videos beyond single frames and decouple segmentation and association.

**Metric Requirements.** Given these insights, we argue for a change of paradigms. In detail, we define the following properties that a STEP metric should satisfy.

- P1. **Analyze full videos at pixel level:** The metric should work on *full videos* and at the *pixel level* rather than on single frames or at segment level.
- P2. **No threshold-based matching:** The metric should treat all pixels equally in space *and* time.
- P3. **No penalty for mistake correction:** Correcting errors in ongoing tracks should not be penalized, but encouraged to obtain long-term track consistency.
- P4. **Consider precision and recall for association:** For association errors (*e.g.*, ID transfer), the metric should take precision and recall into account.
- P5. **Decouple errors:** The metric could be decoupled into components, enabling detailed analysis for different aspects of STEP performance.

## 4.2 STQ Metric

The previous discussions motivate us to propose the Segmentation and Tracking Quality (STQ).

**Formal Definition.** Our benchmark requires a mapping $f$ of every pixel $(x, y, t)$ (indexed by spatial coordinates $(x, y)$ and frame $t$) of a video $\Omega$ to a semantic class $c$ and a track ID $id$. We denote the ground-truth as $gt(x, y, t)$ and the prediction as $pr(x, y, t)$. STQ combines Association Quality (AQ) and Segmentation Quality (SQ) that measure the tracking and segmentation quality, respectively.

**Association Quality (AQ).** The proposed AQ is designed to work at a pixel-level of a full video (*c.f.*, P1). Importantly, all correct and incorrect associations influence the score, independent of whether a segment is above or below an IoU threshold (*c.f.*, P2). We define the prediction for a particular $id$ as:

$$pr_{id}(id) = \{(x, y, t) \mid pr(x, y, t) = (c, id), c \in \mathbf{C}^{th}\} \tag{5}$$

here we only consider "trackable" objects $\mathbf{C}^{th}$, *i.e.*, *things*. The ground-truth is defined analogously. Mislabeling the semantic class of a pixel is *not* penalized in AQ, and will only be judged by the *segmentation quality* described later. This decoupling prevents penalizing wrong semantic predictions twice. We also do not require a consistent semantic class label per track, for reasons illustrated in the following example: A ground-truth van track can easily be mistaken as a car, which should be penalized in the segmentation score, but not in the association score. Moreover, once it becomes clear that the predicted car is actually a van, this class prediction can be corrected which results in an overall increased score for our proposed metric. However, when requiring one semantic class per track, this prediction would be split into one track for car and one for van. As a result, this split that corrects the semantic class would receive a lower score than a prediction that keeps the wrong class, which contradicts P3. We call this effect *class-recovery* in line with the tracking term *ID recovery*.

Following the notation of [43], we define the *true positive associations (TPA)* of an specific ID as:

$$TPA(p, g) = |pr_{id}(p) \cap gt_{id}(g)|. \tag{6}$$

Similarly, *false negative associations (FNA)* and *false positive associations (FPA)* can be defined to compute precision $P_{id}$ and recall $R_{id}$ (*c.f.*, P4). Increasing precision requires the minimization of FPAs, while increasing recall requires minimization of FNAs. We define the $IoU_{id}$ for AQ as follows:

$$IoU_{id}(p, g) = \frac{P_{id}(p, g) \times R_{id}(p, g)}{P_{id}(p, g) + R_{id}(p, g) - P_{id}(p, g) \times R_{id}(p, g)}. \tag{7}$$

Following our goal of long-term track consistency, we encourage *ID recovery* by weighting the score of each predicted tube by their TPA. Without this weighting, a recovered ID would not achieve a

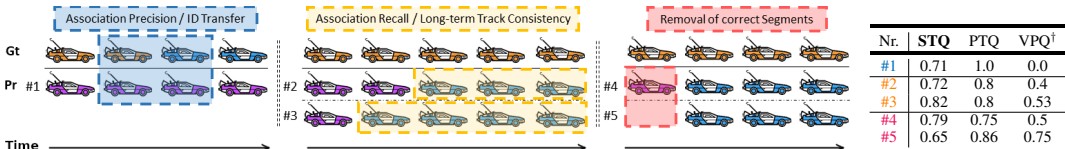

| Nr. | STQ | PTQ | VPQ$^\dagger$ |
|-----|------|------|------|
| #1 | 0.71 | 1.0 | 0.0 |
| #2 | 0.72 | 0.8 | 0.4 |
| #3 | 0.82 | 0.8 | 0.53 |
| #4 | 0.79 | 0.75 | 0.5 |
| #5 | 0.65 | 0.86 | 0.75 |

Figure 5: An illustration of association precision, association recall and the removal of correct segments with wrong track ID for tracks of up to 5 frames. Each car is in a single-frame, where colors encode track IDs. We assume perfect segmentation and show matched tracks. For example, the left scenario contains two ground-truth tracks (orange ■, blue ■), while the prediction contains a single track (violet ■) that overlaps with both ground-truth tracks. Here, only the change of colors is important. Predictions should ideally have color transitions at the same frames as the ground-truth, if any. VPQ$^\dagger$ refers to the VPQ score when evaluated on full videos instead of small spans. STQ is the only metric that properly penalizes ID transfer (#1, P4), encourages long-term track consistency (#3 > #2, P4), and reduces the score when removing semantically correct predictions (#4 > #5, P5).

| Metric Properties | STQ | PTQ [29] | VPQ [34] |
|-------------------|-----|----------|----------|
| P1: Analyze full videos on pixel level | ✓ | ✗ | (✓) |
| P2: No threshold-based matching | ✓ | ✗ | ✗ |
| P3: No penalty for mistake correction | ✓ | ✗ | ✗ |
| P4: Consider precision and recall | ✓ | ✗ | (✓) |
| P5: Decouple errors | ✓ | ✗ | ✗ |

Table 1: Metric comparison. (✓): Partially satisfied. VPQ and PTQ fail to satisfy the properties.

higher score. In total, the association quality (AQ) is defined as follows.

$$AQ = \frac{1}{|\text{gt\_tracks}|} \sum_{g \in \text{gt\_tracks}} \frac{1}{|gt_{id}(g)|} \sum_{p, |p \cap g| \neq \emptyset} TPA(p, g) \times IoU_{id}(p, g) \tag{8}$$

For each ground-truth track $g$, its association score $AQ(g)$ is normalized by its size rather than by the sum of all TPA, which penalizes the removal of correct segments with wrong IDs noticeably.

**Segmentation Quality (SQ).** In semantic segmentation, Intersection-over-Union (IoU) is the most widely adopted metric [21]. As IoU only considers the semantic labels, it fulfills the property of decoupling segmentation and association (*c.f.*, P5). Additionally, it allows us to measure the quality for *crowd* regions[4]. Formally, given $pr(x, y, t)$, and class $c$ we define:

$$pr_{sem}(c) = \{(x, y, t) \mid pr(x, y, t) = (c, *)\}, \tag{9}$$

The ground-truth is defined analogously. We then define SQ to be the mean IoU score:

$$SQ = mIoU = \frac{1}{|\mathbf{C}|} \sum_{c \in \mathbf{C}} \frac{|pr_{sem}(c) \cap gt_{sem}(c)|}{|pr_{sem}(c) \cup gt_{sem}(c)|}, \tag{10}$$

**Segmentation and Tracking Quality (STQ).** The overall STQ score is the geometric mean:

$$STQ = (AQ \times SQ)^{\frac{1}{2}}. \tag{11}$$

The geometric mean is preferable over the arithmetic mean, as it rewards joint segmentation and tracking. Methods specializing in only one aspect of the task will therefore receive a lower score in this setting. The effect of association errors on all proposed metrics is illustrated in Fig. 5 with intermediate computation steps in the supplement. Tab. 1 provides a comparison of STEP metrics. We note that the computational complexity of all metrics is bound by the input size as the $IoU$ computation requires to read all pixels from all frames.

## 5 Baselines

We provide single-frame and multi-frame baselines for the collected STEP datasets. Single-frame baselines follow the *tracking-by-detection* paradigm, *i.e.*, obtaining predictions in each frame independently. Then we associate predicted instances over time to obtain object tracks. Thus, we use separate

---

[4]Regions of *thing* classes with no distinct instances.

| KITTI-STEP | OF | STQ | AQ | SQ | VPQ | PTQ | sPTQ | IDS | sIDS | sMOTSA | MOTSP |
|---|---|---|---|---|---|---|---|---|---|---|---|
| B1: IoU Assoc. | ✗ | 0.58 | 0.47 | **0.71** | **0.44** | *0.48* | 0.48 | 1087 | 914.7 | 0.47 | *0.86* |
| B2: SORT | ✗ | *0.59* | 0.50 | **0.71** | 0.42 | *0.48* | *0.49* | 647 | 536.2 | 0.52 | *0.86* |
| B3: Mask Propagation | ✓ | **0.67** | **0.63** | **0.71** | **0.44** | **0.49** | 0.49 | *533* | *427.4* | *0.54* | *0.86* |
| B4: Motion-DeepLab | ✗ | 0.58 | 0.51 | *0.67* | 0.40 | 0.45 | 0.45 | 659 | 526.7 | 0.44 | 0.84 |
| VPSNet [34] | ✓ | 0.56 | *0.52* | 0.61 | *0.43* | **0.49** | **0.49** | **421** | **360.0** | **0.66** | **0.91** |

Table 2: We compare the different baselines under different metrics on the KITTI-STEP dataset. We highlight the **first** and *second* best score in each metric. OF refers to an external optical flow network.

modules for segmentation and tracking. By contrast, the multi-frame baseline is a *unified* model that jointly tackles segmentation and tracking. Architectural figures are provided in the supplement.

**Base network.** The state-of-art panoptic segmentation model Panoptic-DeepLab [16] is employed as the base network for both single-frame and multi-frame baselines. Panoptic-DeepLab extends DeepLab [11, 12], with an instance segmentation branch that predicts the instance center [66, 68] and regresses every pixel to its center [3, 32, 62]. We use ResNet-50 [26] as the network backbone.

**Single-frame baselines.** Three different methods are used to infer the tracking IDs:

B1. **IoU Association.** The predicted *thing* segments of two consecutive frames are matched, *i.e.*, assigned the same tracking ID by Hungarian Matching [39] with a minimal mask IoU threshold $\delta = 0.3$. To account for occluded objects, unmatched predictions are kept for $\sigma = 10$ frames. Our method is insensitive to $\sigma$, *i.e.*, using $\sigma = 5, 10, 20$ yield similar results.

B2. **SORT Association.** SORT [6] is a simple online tracking method that performs bi-partite matching between sets of track predictions and object detections based on the bounding box overlap criterion. Track predictions are obtained using the Kalman filter. Due to its simplicity, it became a standard baseline for tasks related to tracking [19, 18, 65]. In this paper, we use it as a bounding-box based instance tracking baseline using rectangles that enclose mask predictions.

B3. **Mask Propagation.** We adopt the state-of-art optical flow method RAFT [58, 55] to warp each predicted mask at frame $t - 1$ into frame $t$, followed by the IoU matching (B1). We note that RAFT is highly engineered, trained on multiple datasets and achieves outstanding performance, *e.g.*, 50% error reduction on KITTI [23] compared to FlowNet2 [30]. We also use the forward-backward consistency check [56] to filter out occluded pixels during warping.

**Multi-frame baseline.** Motivated by state-of-the-art MOT methods Tracktor [4] and Center-Track [67], we add another prediction head, *previous-offset*, to the base network that regresses every pixel to its relative instance center in the *previous* frame. Hence, this model can predict motion directly without relying on an external network, which is *only possible due to our dense annotations*, *i.e.*, such multi-frame approaches cannot be trained on sparse data as in Cityscapes-VPS. The previous predicted center heatmap and the previous frame are given as additional inputs to the network:

B4. **Motion-DeepLab.** Using the predicted instance segmentation of the base network, the center of each instance is 'shifted' by the *previous-offset* to find the closest center (within a radius $r$) in the previous frame. We apply a greedy algorithm to match instances in decreasing order of the center's score. Matched instances continue tracks, while unmatched centers start new tracks. Following [67], $r$ equals the geometric mean of the width and height of the predicted instance.

## 6 Results

We showcase the benchmark by studying the performance of different baselines on our datasets through the lens of the proposed STQ metric. In addition to our motion-guided baselines, we evaluate the performance of the state-of-the-art VPS model VPSNet [34]. It uses an optical flow network [30], trained on external densely annotated data to align feature maps from two consecutive frames. Object instances are associated using a trained appearance-based re-id model [65] among other cues.

**Experimental Protocol.** We first pre-train all our baselines without their tracking functionality on Cityscapes [17], and VPSNet on Cityscapes-VPS [34]. These pre-trained networks are then fine-tuned on KITTI-STEP and MOTChallenge-STEP with their tracking functionality enabled. Detailed experimental protocol is shown in the supplement. As MOTChallenge-STEP does not have a validation set, we use sequence 9 for training and 2 for validation.

| MOTChallenge-STEP | OF | STQ | AQ | SQ | VPQ | PTQ | sPTQ | IDS | sIDS | sMOTSA | MOTSP |
|---|---|---|---|---|---|---|---|---|---|---|---|
| B1: IoU Assoc. | ✗ | **0.42** | **0.25** | **0.69** | **0.55** | **0.59** | **0.59** | *164* | **107.6** | 0.21 | 0.77 |
| B2: SORT | ✗ | 0.36 | *0.19* | **0.69** | *0.54* | *0.58* | *0.58* | 364 | 254.5 | 0.22 | 0.77 |
| B3: Mask Propagation | ✓ | *0.41* | **0.25** | **0.69** | **0.55** | *0.58* | **0.59** | 209 | *139.4* | 0.20 | 0.77 |
| B4: Motion-DeepLab | ✗ | 0.35 | *0.19* | *0.62* | 0.51 | 0.54 | 0.54 | 326 | 227.5 | *0.28* | *0.81* |
| VPSNet [34] | ✓ | 0.24 | 0.17 | 0.34 | 0.25 | 0.28 | 0.28 | **152** | 146.3 | **0.40** | **0.84** |

Table 3: Experimental results of different baselines on the MOTChallenge-STEP dataset. We highlight the **first** and *second* best score in each metric. OF refers to an external optical flow network.

**Experimental Results.** In Tab. 2, we report results of all models on KITTI-STEP. To discuss the effects of our STQ metric empirically, we also report scores obtained with VPQ [34] and PTQ [29]. As additional data points, we provide (soft) ID switches (s)IDS, sMOTSA and MOTS Precision (MOTSP) [61] scores that, in contrast to the other metrics, only evaluate 'cars' and 'pedestrians'. We observe that our single-frame baselines (*B1-B3*) that perform panoptic segmentation and tracking separately achieve the overall highest STQ scores. When using multiple separate state-of-the-art models (one for panoptic segmentation and the one for optical flow), *B3* achieves the highest performance in terms of association quality (AQ) and also overall in STQ. Both multi-frame baselines, *B4* and *VPSNet* are tackling a significantly more challenging problem, addressing segmentation and tracking jointly within the network. This comes at the cost of a decrease in a single-task performance (*c.f.* SQ), which is also observed in other multi-task settings [37] such as image panoptic segmentation [36]. Initially, separate models outperformed unified models, taking significant research effort into unified models to catch up. *B4* is such a unified model that does not rely on any external network and tackles panoptic segmentation and tracking jointly. When studying the effect of motion cues (*B4*) and appearance re-id cues (*VPSNet*), we notice that both significantly impact the tracking performance, as can be seen from the improvement in terms of AQ over *B1*.

From a tracking perspective, MOTChallenge-STEP is more challenging compared to KITTI-STEP because it contains several overlapping pedestrian tracks and a reduced amount of training data. In Tab. 3, we observe that *B1* and *B3* achieve a similar performance, which we attribute to the reduced inter-frame motion coming from pedestrians and the high 30 FPS frame rate. Notably, the *SORT* tracking fails to achieve track consistency in the scenarios of many overlapping instances. Naturally, separate segmentation and tracking models (*B1-B3*) are less affected by the reduced amount of training data, and therefore achieve consistently better results compared to the multi-frame baselines. However, the unified tracking and segmentation models *B4* and *VPSNet*, need to train their tracking heads on reduced data and therefore easily run into overfitting issues as is evident from reduced scores. Specifically, *VPSNet* achieves a low SQ score, and therefore a low recall yet a high precision (*c.f.* sMOTSA, MOTSP). In contrast, *B4* does better in SQ, but cannot leverage this to improve tracking.

In Tab. 2 and Tab. 3, we also show the effect of different metrics. We can experimentally confirm our findings from the metric analysis (Sec. 4). When comparing our metric STQ with VPQ and PTQ, we observe that even though tracking quality varies significantly (*c.f.* AQ, IDS, sMOTSA), the changes in VPQ and PTQ are rarely noticeable. This supports our theoretical findings that VPQ and PTQ evaluate this task from a segmentation perspective, leading to high scores for *B1-B4* in Tab. 3, and fail to accurately reflect changes in association. This supports the need for a new perspective on this task, STEP, which equally emphasizes tracking and segmentation.

In summary, our experimental findings show that the novel STQ metric captures the essence of segmentation and tracking by considering both aspects equally. As our datasets KITTI-STEP and MOTChallenge-STEP provide dense annotations, for the first time motion and appearance cues can be leveraged in a unified model for dense, pixel-precise video scene understanding.

## 7 Conclusion

In this paper, we present a new perspective on the task of video panoptic segmentation. We provide a new benchmark, Segmenting and Tracking Every Pixel (STEP), to the community where we explicitly focus on measuring algorithm performance at the most detailed level possible, taking each pixel into account. For that purpose, we analyze the drawbacks of existing metrics, and propose the STQ metric. Our benchmark and metric are designed for evaluating algorithms in real-world scenarios, where understanding long-term tracking performance is important. We believe that this work provides an important STEP towards a dense, pixel-precise video understanding.

**Acknowledgments.** We would like to thank Sérgio Agostinho, Deqing Sun and Siyuan Qiao for their valuable feedback, and Stefan Popov and Vittorio Ferrari for their work on the annotation tool. We would also like to thank for the supports from Google AI:D, Mobile Vision, the TUM Dynamic Vision and Learning Group, and all the Crowd Compute raters. This work has been partially funded by the German Federal Ministry of Education and Research (BMBF) under Grant No. 01IS18036B. The authors of this work take full responsibility for its content.

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
