# OpenReview forum: "STEP: Segmenting and Tracking Every Pixel"
_NeurIPS.cc/2021/Track/Datasets_and_Benchmarks/Round2 — NeurIPS 2021 Datasets and Benchmarks Track (Round 2)_

### Official Review · Reviewer_4XNi · 2021-09-16
**Two datasets for video panoptic segmentation, but with less new insights introduced**

**Rating:** 5
**Confidence:** 3
**Correctness:** The claims seem to be correct.
**Clarity:** Yes.

**Strengths:**

This paper introduces two datasets for video panoptic segmentation by providing extra annotations for more semantic classes. I appreciate this effort.

The authors investigate several metrics and develop a new one for this task.

Experimental results are shown for a few baselines with analysis. This can provide comparisons for future research.

**Weaknesses:**

My biggest concern with this work is its major contribution, i.e., the two proposed benchmarks KITTI-STEP and MOTChallenge-STEP. There are some questions:

(1) how do these two benchmarks help improve the understanding of scenes in videos? In addition to adding some more annotation, I don't see much big difference from existing datasets. In addition, in these extra annotations, some are actually not necessary to track.

(2) what do we learn from Fig. 1? I don't see any indication in the paper to Fig. 1. Is it the result of baseline? Or do you want to show the annotation in your datasets?

(3) From my understanding, Fig. 1 is actually similar to video semantic segmentation. Is there anything related to tracking each pixel in the video?

(4) The statement "tracking every pixel" is confusing (at least to me). When we talk about tracking something, we usually assign an identity to that object. For example, tracking single object means we need to assign a unique label to the target of interest. Likewise, tracking multiple objects needs to assign IDs to different objects. When I see tracking pixels, I will think about assigning IDs to each pixels regardless of their semantic categories. Correct me if this is a common way to call the task in your field.

(5) How do you plan to explore temporal information in the videos with average length 51 (KITTI-STEP) and 187 (MOTChallenge-STEP), especially considering there exists information redundancy in these videos?


**Additional Feedback:**

Please check out the weakness part.

**Documentation:**

The documentation is good.

**Ethics:**

I didn't see any ethical problems.

**Relation To Prior Work:**

The authors provide rich discussions with previous works.

**Summary And Contributions:**

This paper introduces two datasets, KITTI-STEP and MOTChallenge-STEP, for the task of video panoptic (segmenting and tracking pixels in a video) by extending two existing datasets KITTI and MOTChallenge. To improve over existing benchmarks, this paper introduces extra annotations by labeling more semantic classes from the background. Compared with previous benchmarks, the proposed contain more dense and long-term annotations. Further, the authors also investigate the metrics for this task and present a new metric by providing an accurate and intuitive comparison against the GT at a fine-grained level. In the final part, experiments are conducted for some baselines with analysis.

---

> ### Author Response · Authors · 2021-09-28
> **Responds to reviewer #4 - part 1**
>
> We thank the reviewer for their review! We are glad the reviewer appreciated the significant effort to annotate the proposed datasets and found the proposed metric to be accurate and intuitive. We appreciate the detailed questions about the dataset and the opportunity to clarify differences over existing work.
>
> ---
> ***“How do these benchmarks help improve the understanding of videos? What are the improvements over existing datasets?”***
>
> In video panoptic segmentation, the task is to assign a semantic label and an instance ID to every pixel. This task goes beyond the requirement for tracking benchmarks. The task of video panoptic segmentation helps to improve the understanding of videos compared to the task of tracking by requiring a more **holistic scene understanding**. We humans process continuous streams of video data all the time; we can reason about instances and future motion as well as the semantic meaning of observed surroundings and surfaces. Eventually, robots (will) require similar abilities. So far, there exist no public real-world dataset that can be used for benchmarking such settings.
>
> - Improvements over KITTI and MOTChallenge: The challenges introduced by the new benchmarks require that beyond tracking, every pixel has to be segmented. In the original KITTI and MOTChallenge datasets, the vast majority of pixels are unlabelled as the focus is on pedestrians and cars. However, having a complete understanding of surroundings helps in many down-stream applications as discussed before. One of many examples is: Being able to precisely segment road/drivable areas is important for planning motion and path-finding.
> - Improvements over Cityscapes-VPS: For a quantitative and detailed comparison, we refer to the dataset comparison in the main paper and Sec. C of the supplement. One key difference is the duration of videos and therefore the ability to properly evaluate tracking. In the proposed benchmarks tracking performance can be evaluated for long sequences (some longer than a minute), while in Cityscapes-VPS videos are 1.8 seconds long. Moreover, their sparse ground-truth means that instances are on average only visible for 3.3 annotated frames, which puts a strong focus on the segmentation. In most robotic scenarios, there's a requirement to be able to track objects consistently and for a long time. Due to the extensive labelling process, prior work has focussed on short clips. Our goal for this benchmark is to provide a real-world test-bed for algorithms to evaluate both their segmentation as well as their tracking performance down to each single pixel.
>
> ***“Some of these new annotations are not necessary for tracking”***
>
> Indeed, prior research has shown that multi-object tracking methods do not require semantic labels for background regions. However, **the proposed benchmarks are not (just) tracking but segmentation and tracking benchmarks**. To be able to measure the segmentation performance of such methods, accurate segmentation ground-truth for every pixel is required. Hence, the **new annotations are necessary to evaluate the task**. As a minor remark, one could argue that temporally coherent semantic interpretation of a scene will implicitly lead to the emergence of some form of tracking (for stuff classes), but that is not the main point why the new annotations are necessary.
>
> \
> **“What does Figure 1 show?”**
>
> Our teaser image, **Figure 1, shows ground-truth annotations** from proposed datasets. We have clarified that in the caption and referenced it in the manuscript.
>
> \
> ***“Does Figure 1 only show semantic annotations?”***
>
> **Not quite**, the labels are showing segmentation classes (different colors) and each track has a different shade of the segmentation class base color. We appreciate the reviewer’s comment as it makes us realize that the small color differences between different tracks are too subtle. **We are working on a visualisation with higher contrast** to make it more clear that not only semantic classes are visualized.

---

> > ### Author Response · Authors · 2021-09-28
> > **Response to reviewer #4 - part 2**
> >
> > ***“Prior tasks usually assign an identity to one or multiple objects. When I see tracking pixels, I think of assigning IDs to each pixel regardless of their semantic category.”***
> >
> > The reviewer’s understanding makes a good point about a new feature of the proposed pixel-level evaluation.
> > In MOT(S), the reviewer's description is accurate. A **single** ID is assigned to an object. Objects can be bounding boxes or masks.
> > **In our task, tracking IDs are not assigned on object-level but on pixel-level**. And indeed, **tracking IDs and semantic classes are independent in the proposed STQ metric**. This allows predictions to have different semantic categories for pixels that all share the same tracking ID. For a method doing online tracking, this allows for correcting semantic predictions.
> >
> > Considering the following example: A distant car is driving towards the camera. At first, an online method might confuse the car with a van, thus predicting the wrong class. Then, in close range, the car can be confidently categorized as a car. Yet, the method knows it's still the same track. For these cases, allowing one track to contain more than a single predicted semantic class encourages online methods to correct mistakes without getting penalized in the tracking score. A penalty however is applied in the semantic measure. Requiring a single semantic class and tracking ID on object-level would mean that in the above example, the predicted track is split into a van track and a car track, even though the method could say it is the same track. More details to this and other examples, and how this influenced the design of the proposed STQ metric can be found in Sec. E in the supplement.
> >
> > \
> > ***“How do you plan to explore temporal information coming from long videos, especially considering redundancy?”***
> >
> > We interpret this question as how can the available temporal information be leveraged for developing methods in future work. This is an interesting and open question. Research on video classification and action recognition has shown that many frames can be encoded together. Recent methods in Video Instance Segmentation and LiDAR Point Cloud Segmentation already increase the temporal window used as (single) input to several frames successfully. Orthogonal to this, there is research on using gating functions in video segmentation to partially propagate information from previous frames in case of redundancy.
> >
> > However, in many real-world robotic scenarios such as self-driving cars, it is not possible to just simply skip chunks of frames due to potential redundancy as there might occur safety-critical changes that need to be observed. We are excited for future work to pick up on this reviewer’s observation and improve upon our baseline.
> >
> > In case the interpretation of the question is not correct, we would appreciate if the reviewer could clarify the question.

---

### Official Review · Reviewer_yzVm · 2021-09-17
**This work proposes new datasets and evaluation metric for the video panoptic segmentation task.**

**Rating:** 5
**Confidence:** 5
**Correctness:** Yes

**Strengths:**

- The KITTI-STEP and MOTChallenge-STEP datasets are meaningful due to their high-frame and long-duration annotations.
- The challenges based on the proposed dataset are indeed contributions to the research of the related fields.

**Weaknesses:**

I appreciate the efforts by the authors to create the datasets and organize the challenges. However, I have several concerns about this paper.

- The task definition that "Tracking Every Pixel" is somehow overclaimed to me. I understand that the authors may mean that their evaluation considers every pixel. Yet, "Tracking Every Pixel" always reminds me tasks like optical/scene flows that define identities for each pixel. This paper just tracks at the instance level, and is exactly a video panoptic segmentation task. Hence, it is improper to overly emphasize "every pixel" in this paper.

- In the first item of  "Contributions", the authors claim that the proposed datasets are "more suitable" for Spatio-temporal dense tasks. This is not very convincing to me, as in the proposed dataset only two classes ("Pedestrian" and "Car") are considered as instances, which ignores many other traffic participants as defined in CItyscapes. Moreover, there also lacks analysis about the quality of the annotations.

- I also noticed that the STQ has been proposed in [a]. Though [a] is for 3D LiDAR data, the very similar formulation makes STQ hard to be a contribution of this work.

- The content of  Ln63-69 is overlapped with Ln46-62.

- It would be better to add some qualitative analysis about the datasets.

[a] 4D Panoptic LiDAR Segmentation, CVPR21


**Additional Feedback:**

No

**Clarity:**

I believe that it is improper to overly emphasize "every pixel" in this paper, since
- All the previous evaluation metrics are computed from pixels.
- The proposed STQ actually measures at the instance level, and its advantages for pixel-level quality are not obvious.

**Documentation:**

Yes

**Relation To Prior Work:**

Yes

**Summary And Contributions:**

This paper presents two datasets for long-term video panoptic segmentation. The datasets are annotated with high frame rates and long durations. In addition, the authors also propose the Segmentation and Tracking Quality (STQ) for better evaluating the performance of VPS models.

---

> ### Author Response · Authors · 2021-09-28
> **Responds to reviewer #3 - part 1**
>
> We thank the reviewer for their review! We are grateful for the reviewer’s feedback and are happy to clarify some misunderstandings. We agree with the reviewer that the proposed dataset annotations “are meaningful due their high frame-rate and long-duration”, offering a novel test-bed for real-world evaluation. We are pleased that the reviewer recognizes the benchmark/challenges based on the proposed dataset as “contributions to the research of the related fields”.
>
> ---
> ***“The proposed metric STQ is not a contribution of this work as it has been used in published work [1].”***
>
> **No**, this is a misunderstanding. **The proposed metric STQ is a major contribution of this work.** Even though this work is not yet published, our proposed metric is already adopted for a set of other benchmarks [1, 2b, 3], and all of them give credit to an earlier released preprint of this work. The focus of [1] was on the extension of this task to the LiDAR domain and the proposition of a new method. The work of [1] summarizes the STQ metric and applies it to the LiDAR domain. In fact, the following quote comes from the work [1] the reviewer mentioned:
>
> > Inspired by [...] the concurrent work on video panoptic segmentation proposing the
> Segmentation and Tracking Quality (STQ) [75], our LSTQ (LiDAR Segmentation and Tracking Quality) consists of two terms [...]. (Sec. 3.2.2, page 5, [1])
>
> We view this as a clear attribution to our work. Besides the work of [1], the published benchmark [2b] from the CVPR2021 paper [2a] has also adopted the proposed metric by extending it with a depth measure. Furthermore, a recently released preprint [3], heavily builds on our metric analysis. That preprint also adopts the proposed association quality that is part of STQ but changes it from a pixel/point-level evaluation to an instance-level evaluation. In all these cases, a clear attribution is given to our work and we are pleased that this work has already inspired others showing the relevance of the proposed metric beyond our benchmark.
>
>
> [1] 4D Panoptic LiDAR Segmentation, CVPR21
>
> [2a] ViP-DeepLab: Learning Visual Perception with Depth-aware Video Panoptic Segmentation, CVPR21
>
> [2b] SemKITTI-DVPS: https://competitions.codalab.org/competitions/33634
>
> [3] Panoptic nuScenes: A Large-Scale Benchmark for LiDAR Panoptic Segmentation and Tracking, arXiv preprint
>
> \
> ***“The proposed metric STQ actually measures at the instance-level.”***
>
> **No**, this is a misunderstanding as STQ measures at **pixel-level**. We hope that the following description makes the difference more clear:
>
> We consider the prior metric PTQ as an example for instance-level evaluation of tracking. To obtain the set of true positives (TP), false positives (FP) and false negatives (FN), the predictions are compared with the ground-truth and a binary decision is made whether a whole predicted segment belongs to the set of TP or FP. This binary decision is based on an IoU threshold of 0.5. Above the threshold means the complete segment is a single true positive, below or equal means a single false positive. Based on these sets, the ID switches are computed, thus computing the tracking evaluation on instance-level.
>
> In contrast, in STQ the set of true positives for association (TPA), and likewise the FPA and FNA, are collected on pixel-level. This means these sets are collections of pixels and not of segments/instances. Hence, STQ measures both segmentation and tracking at pixel-level. By avoiding the binary decision, every pixel has an influence on the association score.
>
> Please also note the answer to a similar question from another reviewer.
>
> \
> ***“It would be better to add some qualitative analysis about the dataset.”***
>
> If we understand the recommendation correctly, this has been part of the **supplement**. As it is very hard to judge the annotations from single images (c.f. Figure 1 main paper), we believe that video annotations are best viewed as videos. Due to the very strict file size requirements of this review platform, we have linked the aforementioned video in the supplement like we also linked the code, challenges and datasets.
>
> In case the reviewer believes that more images showing annotations like in Figure 1 are helpful, we are very happy to add these to the supplement showing the proposed dataset annotations and comparing it to related benchmarks.

---

> > ### Author Response · Authors · 2021-09-28
> > **Responds to reviewer #3 - part 2**
> >
> > ***“What are the advantages of pixel-level quality measures?”***
> >
> > **A pixel-level evaluation offers a richer granularity of the measurement compared to instance-level evaluation.** This is why the paper points out that every pixel has an influence on the association score. An intuitive example is described in the paper for PQ (c.f. Sec. 4.1): We consider the following example, a human is shown two images. In both, a pair of ground-truth and prediction is shown. In one image, the prediction and ground-truth are aligned such that the IoU is 0.49. In the other image, the same pair is aligned such that the IoU is 0.51. Since the visual difference in both images is so small (it could be just a few pixel difference), it can be hardly recognized with human eyes. Moreover, it is not intuitive that one case is regarded as the object is successfully detected and is part of a track while the other case is regarded as the object is not detected at all and therefore not part of a track (c.f. answer above). After all, both cases look quite similar to humans, hence we propose a measure that penalizes such small visual differences only with a small difference in score.
> >
> > In our proposed STQ metric, all predictions will be taken into account weighted according to their overlap. We argue this aligns better with the human judgement of the quality of predictions, while still ensuring that the case of IoU 0.51 achieves a better score than the case of IoU 0.49.
> >
> > An example visualisation of this can be found here:
> > [link](https://drive.google.com/file/d/1d4CqIkKvR136JUcqpQ4-HGiLm-2eLqDO/view?usp=sharing)
> >
> > \
> > ***“The proposed benchmark is not ‘more suitable’ for spatio-temporal dense tasks.”***
> >
> > We disagree with that assessment. We note that the previously only existing real-world dataset for this task (Cityscapes-VPS) has made a substantial contribution to this research field. In this work, we aim to address some of the shortcomings of Cityscapes-VPS. We would like to support our view that our proposed benchmark is more suitable with the following:
> > - A “spatio-temporal dense task” is evaluated best at a spatio-temporal dense level. This requires spatio-temporal dense annotations and a dense evaluation measure. Cityscapes-VPS is labelled sparsely (every 5th frame) due to the expensive labelling process. In our proposed datasets, we have labelled every frame enabling for the first time spatio-temporal dense evaluation with our proposed metric on real-world data for this task.
> > - A “spatio-temporal dense task” that requires tracking benefits from larger temporal extensions of the input, as this comes closer to scenarios in which continuous streams of data need to be interpreted such as in autonomous driving. In Cityscapes-VPS, all clips have a length of 1.8 seconds while our datasets have videos up to 100 seconds long.
> > - We argue the proposed benchmark has a higher difficulty than the existing one. Our proposed benchmark inherits the already challenging semantic interpretation in Cityscapes-VPS with 19 classes by adopting the same number of classes. On top of this, our proposed benchmark adds additional difficulty in tracking. We refer to the tracking difficulty discussion in Sec. 3 and have dedicated a whole section in the supplement to this discussion (Sec. C). In summary:
> > 1. In Cityscapes-VPS, close to half (47%) of all instances in the validation set  are only visible in a single annotated frame, requiring no tracking at all. This is an effect of the sparse labelling.
> > 2. Tracking difficulty scales with track length and occlusions. The average track length in Cityscapes-VPS validation set is 3.3 annotated frames, while for example in the proposed KITTI-STEP it is 50 annotated frames.
> > 3. We believe that the number of tracking classes is not the best metric when comparing tracking difficulty between datasets. As an example, the Cityscapes-VPS has a tracking class train, yet has only a single train track in the whole validation set. Thus, only requiring semantic interpretation in that case.
> >
> > \
> > ***“It is improper to overly emphasize every pixel.”***
> >
> > We hope with the clarifications provided in the question on the metric it becomes more clear why we point this out. As a summary:
> >
> > - The task of video panoptic segmentation is a task that requires predictions for every pixel.
> > - In this work, we propose a metric that works on the pixel-level and in which every pixel has an influence on the association/tracking score.
> > - We propose spatio-temporal dense annotations that contain labels for every pixel.
> >
> > Considering all this, we believe it is fair to emphasize that now the task definition, the metric and the datasets all consider every pixel.

---

### Official Review · Reviewer_369f · 2021-09-19

**Rating:** 7
**Confidence:** 3
**Correctness:** The metric seems correct and it is ba…
**Clarity:** The paper is very clear.

**Strengths:**

-	The paper is a pleasure to read and easy to follow.
-	The limitations of the previous data set and metrics are carefully studied and explained.
-	The design of the new STQ metric is very well explained.


**Weaknesses:**

-	In the main manuscript, there is limited information about how the data set was collected and more visual examples should be added.

**Additional Feedback:**

-	In the main manuscript, there is limited information about how the data set was collected and more visual examples should be added.

**Documentation:**

yes

**Ethics:**

no concerns.

**Relation To Prior Work:**

The related work is carefully studied.

**Summary And Contributions:**

This paper presents a new dataset and metric for segmentation and tracking. The old metrics for segmentation and tracking are carefully study and their limitations are explained. The new proposed metric takes into account the segmentation and tracking performance without preferring one over the other.

---

> ### Author Response · Authors · 2021-09-28
> **Response to reviewer #2**
>
> We thank the reviewer for their review! We are pleased by the reviewer’s acknowledgement of:
> - **Novelty / Relevance**: “New dataset”, “new metric”
> - **Thoroughness**: “old metrics are carefully studied and their limitations are explained”
> - **Clarity**: “The paper is a pleasure to read an easy to follow”, “The design of the new STQ metric is very well explained”
>
> ---
> ***“Could the authors add more information and visual examples from the supplement to the main paper about dataset collection?”***
>
> **Yes**, we are happy to do so. NeurIPS allows an additional page of content for the camera-ready version. Hence, we **extend** the dataset section of the main paper. We have **added**:
> - Figure 1 from the supplement to the main paper. This figure shows the **semantic class distribution** for both proposed datasets.
> - Figure 3 from the supplement to the main paper. This figure illustrates the annotation process by giving **real visual examples** of how the annotations change throughout this process. It shows the multi-step approach we took for labelling the proposed datasets. First, a network is used to predict labels. Second, a human annotator carefully refines and corrects the annotation in multiple passes to achieve high-quality results. For different passes, different human annotators were refining the labels. The final step is the merging procedure with the instance labels.
> - Text explaining the added figures.
>
> We are happy to add more details, if the reviewer feels that some other information should be included as well.
>
> We will upload a revised version soon.

---

### Official Review · Reviewer_pZRb · 2021-09-21
**Very nice paper, I would like to give a high rating.**

**Rating:** 9
**Confidence:** 4
**Correctness:** Yes
**Clarity:** Yes, the paper is well written.

**Strengths:**

detailed and easy-to-follow descriptions including the processes of data collection and annotation, metric computation and comparison, baseline implementation, and result analysis.

**Weaknesses:**

- Although the comparison of evaluation metrics is described to show the benefits of the proposed metric, I think the computational complexity of each metric is worth analyzing.
- Figure 3 is not easy to understand to me. Are the colors of gt cars correct? And the color masks may not necessary.

**Additional Feedback:**

Please see the weakness.

**Documentation:**

Yes, I think it is sufficient.

**Ethics:**

No.

**Relation To Prior Work:**

Yes, it is clear.

**Summary And Contributions:**

This paper proposed datasets, a new evaluation metric, and baselines for video panoptic segmentation. I believe this work will promote the development of this field. Their contributions include:

- based on the KITTI dataset, the authors created KITTI-STEP and MOTChalleng-STEP datasets with pixel-level annotations for every frame.
- proposed a new metric to evaluate the task of video panoptic segmentation.
- introduced baselines and tested them on the created dataset to show the challenges.

---

> ### Author Response · Authors · 2021-09-28
> **Response to reviewer #1**
>
> We thank the reviewer for their review! We are encouraged and glad the reviewer found:
> - **Impact / Relevance**: “this work will promote the development of this field.”
> - **Completeness**: “detailed and easy-to-follow descriptions including the processes of data collection and annotation, metric computation and comparison, baseline implementation and result analysis.”
> - **Clarity**: “The paper is well written.”, “detailed and easy-to-follow”
>
> ---
> ***“Could the authors add a computation complexity analysis for the discussed metrics?”***
>
> **We are happy to provide such analysis in the paper.** We would like to ask the reviewer for a clarification on this. Does the question refer to time measurements or on analysis on the runtime complexity (Big-O notation)? We would like to point out that in all metrics, the processing of all pixels for the sake of computing IoU-scores dominates all other computation steps.
>
> \
> ***“Are the ground-truth colors in Figure 3 correct?”***
>
> **Yes**, the colors are correct. Please note that colors only encode track IDs not “matches”. The first example has four frames and contains two ground-truth tracks (one orange, and one blue). The prediction contains a single track (purple). The scenario assumes perfect spatial overlap (segmentation) for simplicity. We have highlighted the mistake (“ID Transfer”) with a blue box. For a perfect score, the prediction should have a color change at the same frame as the ground-truth since this would mean the prediction has recognized the two tracks.
>
> In summary, only **changes in color** are important, not which colors the tracks have. If the ground-truth has no change in color (single track, c.f. #2 and #3), the prediction should also contain no change. If the ground-truth has a color change (multiple tracks, c.f. #1), the prediction should have color changes at the same frames.
>
> **We add the following sentence to the caption of Figure 3**:
> > For example, the left scenario contains two ground-truth tracks (orange, blue), while the prediction contains a single track (purple) that overlaps with both ground-truth tracks. Here, only the change of colors is important. Predictions should ideally have color transitions at the same frames as the ground-truth, if any.
>
> In case the confusion is caused by color blindness, we would very much appreciate any comment on this, so we can try different colors.
>
> \
> ***“Are the color masks in Figure 3 necessary?”***
>
> We would like to ask the reviewer whether the above answer already answers this question, too. If not, we would like to ask the reviewer to rephrase the question so we can further improve this figure.

---

> > ### Comment · Reviewer_pZRb · 2021-09-29
> > **Further discussion**
> >
> > - Does the question refer to time measurements or on analysis on the runtime complexity (Big-O notation)?
> >
> > Yes.
> >
> > - In case the confusion is caused by color blindness, we would very much appreciate any comment on this, so we can try different colors.
> >
> > Thank you. You can indicate the color in the caption at least.

---

> > > ### Author Response · Authors · 2021-09-29
> > > **Complexity**
> > >
> > > Thank you for your response.
> > >
> > >
> > > ***Runtime Complexity***
> > >
> > > Let
> > > - $m$ be the number of pixels per frame.
> > > - $f$ be the number of frames.
> > > - $k$ be the hyper-parameter of VPQ. Please note $k \leq f$.
> > > - $p$ be the number of predicted tracks.
> > > - $g$ be the number of ground-truth tracks.
> > > - $c$ be the number of classes.
> > >
> > > It holds that $\max(p, g) \leq mf$, since at most every pixel can be a new track.
> > >
> > > | Metric | $IoU$ | $TP_c + \frac{1}{2}FP_c + \frac{1}{2}FN_c$ | $AQ$ | Overall |
> > > |:--------:|:--------:|:------------:|:--------:|:----------:|
> > > | PTQ   | $\mathcal{O}(mf)$    |  $\mathcal{O}(\max(p, g)mf) = \mathcal{O}(mf)$       | n/a | $\mathcal{O}(cmf)$  |
> > > | VPQ   | $\mathcal{O}(mfk) = \mathcal{O}(mf)$    | $\mathcal{O}(\max(p, g)kmf) = \mathcal{O}(mf)$          | n/a | $\mathcal{O}(cmf)$  |
> > > | STQ   | $ \mathcal{O}(mf)$    | n/a        |  $\mathcal{O}(\max(p, g)mf) = \mathcal{O}(mf)$ | $\mathcal{O}(cmf)$  |
> > >
> > > We note, that usually $c \ll mf$.
> > >
> > > In all these cases, the runtime complexity is dominated by the requirement of reading the input, $m$ pixels by $f$ frames. All further computation steps have a lower runtime complexity. Please note, in practice for all the metrics, the input is read once and then directly used to compute IoU scores in an efficient way. Hence, if the $IoU$ score is computed once, it can be reused for $TP_c+ \frac{1}{2}FP_c + \frac{1}{2}FN_c$ and $AQ$. Besides the theoretical complexity, we have observed that the runtime heavily depends on implementation details like for example:
> > >
> > > - Which image loader is used to read images.
> > > - How many divisions and modulo operations are used. We observe large runtime improvements by using bit-masking and bit-shifting instead.
> > >
> > > ***Figure 3.***
> > >
> > > Thank you very much for your recommendation. We will add this.

---

### Author Response · Authors · 2021-09-29
**Revised paper**

We would like to thank all reviewers again for their feedback! Please see our detailed response below each review.


**Changelog:**

- We clarified Figure 1 caption and increased the contrast on instances for the label only visualisation (right images). We reference it also now in the paper.
- We have added Figure 2 (supplement before) detailing the dataset collection process as well as how the quality evolves from machine annotation over multiple passes of human refinement.
- We have added Figure 4 (supplement before) showing the semantic class distribution and giving insights into the most salient classes.
- We have clarified the caption of Figure 5 (Figure 3 before) with the suggestions of reviewer 1. We have indicated all colors in the caption and have given more information on how to read the figure.
- We have added a few sentences about the computational complexity of all metrics which is bound by the input size as every pixel needs to be read.

---

### Decision · Program_Chairs · 2021-10-09

**Decision:**

Accept

**Comment:**

The paper received somewhat diverging ratings, from strong accept to weak reject. The AC read the paper, the reviews and the authors' responses. I do understand the reason why some concerns were raised by reviewers about what it means to track every pixel and other such nuances, but I do find myself to side with the authors. The dataset is interesting and opens the door for more comprehensive and challenging visual recognition. The paper is well written and well executed. So, congratulations to authors!!